# Phenolic Compounds and Antioxidant Capacity Comparison of Wild-Type and Yellow-Leaf *gl1* Mutant of *Lagerstroemia indica*

**DOI:** 10.3390/plants13020315

**Published:** 2024-01-20

**Authors:** Sumei Li, Min Yin, Peng Wang, Lulu Gao, Fenni Lv, Rutong Yang, Ya Li, Qing Wang, Linfang Li, Yongdong Liu, Shuan Wang

**Affiliations:** Jiangsu Key Laboratory for the Research and Utilization of Plant Resources, Institute of Botany, Jiangsu Province and Chinese Academy of Sciences, Nanjing Botanical Garden, Memorial Sun Yat-Sen, No. 1 Qianhu Houcun, Nanjing 210014, China; smli321@163.com (S.L.); epmin@sohu.com (M.Y.); wp280018@163.com (P.W.); gaolulu@jib.ac.cn (L.G.); fennilv@cnbg.net (F.L.); yangrt2000@126.com (R.Y.); yalicnbg@163.com (Y.L.); wq926cn@hotmail.com (Q.W.); lilinfangqq@163.com (L.L.); liuyongdong@jib.ac.cn (Y.L.)

**Keywords:** phenolic compounds, antioxidant activity, metabolomics, yellow-leaf mutant, *Lagerstroemia indica*

## Abstract

Background: The yellow-leaf *gl1* mutant of *Lagerstroemia indica* exhibits an altered phenylpropanoid metabolism pathway compared to wild-type (WT). However, details on the metabolites associated with leaf color variation, including color-specific metabolites with bioactive constituents, are not fully understood. Methods: Chemical and metabolomics approaches were used to compare metabolite composition and antioxidant capacity between the *gl1* mutant and WT leaves. Results: The mutant exhibited an irregular xylem structure with a significantly lower phenolic polymer lignin content and higher soluble phenolic compounds. Untargeted metabolomics analysis identified phenolic compounds, particularly lignans, as key differential metabolites between *gl1* and WT, with a significant increase in the mutant. The neolignan derivative balanophonin-4-*O*-D-glu was identified as a characteristic metabolite in the *gl1* mutant. The soluble phenolic compounds of the *gl1* mutant exhibited higher FRAP, ABTS, DPPH, and hydroxyl radical scavenging activity than in WT. Correlation analysis showed a positive relationship between antioxidant capacity and phenolic compounds in *L. indica*. Conclusions: Metabolites associated with leaf color variation in the *L. indica* yellow-leaf *gl1* mutant demonstrated high antioxidant capacity, particularly in scavenging hydroxyl radicals.

## 1. Introduction

The genus *Lagerstroemia* L., belonging to the family Lythraceae and order Myrtales, includes over 50 species of deciduous and evergreen trees and shrubs, widespread throughout Southeast Asia and Australia [1,2]. *Lagerstroemia* species mitigate symptoms of many serious diseases and health problems via bioactive processes by acting as antioxidants [3,4,5,6], anti-inflammatory [4,7], anti-aging [2], anti-obesity, and antidiabetic compounds [8,9,10]. The leaves of *Lagerstroemia speciosa* are used as traditional medicinal foods in Southeast Asia, including in the Philippines, Vietnam, Malaysia, and southern China [11]. Phytochemical investigations have demonstrated that *Lagerstroemia* contains diverse bioactive metabolites, including phenolic compounds [9,11,12,13,14,15,16,17], terpenoids [9,12,17], and alkaloids [18,19]. As a result, the bark, leaves, flowers, and fruits of the plants have been analyzed to demonstrate the possibility of using *Lagerstroemia* species as a source of bioactive compounds for the pharmacological industry. The major active ingredients within *Lagerstroemia* species are triterpene acids such as corosolic acid and asiatic acid [9,10]. However, the phenolic compounds, including tannin, ellagic acid, myricetin, quercetin, and their derivates, are responsible for the various pharmacological functions in *Lagerstroemia* species, such as anti-aging potentials [2], antioxidative acitivity [3], and anti-human rhinovirus 2 activity [11].

The health benefits of plant products, including disease prevention, are often attributed to their high content of phenolic compounds [20,21]. Phenolic compounds are among the most widespread secondary metabolites, ubiquitously present throughout the plant kingdom. Phenylalanine is generated in the shikimate pathway, leading to phenylpropanoid biosynthesis. Land plants have evolved multiple branches of phenylpropanoid metabolism, giving rise to more than 8000 metabolites with an extremely wide variation in structural diversity exhibiting a broad range of physiological roles in plants [22]. They can be classified into several classes based on the variations in their basic skeletons, including phenolic acids, lignans, lignins, flavonoids, stilbenes, anthraquinones, and others [23]. Phenylpropanoid metabolism is regulated via the redirection of metabolic flux, which exhibits extraordinary complexity and plasticity homeostasis among different branches in response to the changing environment [24]. Phenylpropanoid metabolism is, therefore, one of the most extensively investigated pathways [22]. Phenolic compounds are thus the characteristic bioactive compounds in plants due to their importance in the various interactions between plants and their environment, reproductive strategies, and defense mechanisms [25].

*Lagerstroemia indica* originated and has extensive cultivation history in China [26]. The extract of *L. indica* can alleviate several symptoms, including stretch marks [27] and allergic airway inflammation [7]. Based on the phytochemical profiling of bioactive metabolites, this species is clustered into the same group with *L. speciosa* among 17 different *Lagerstroemia* species [9]. The yellow-leaf mutant *gl1* is cultivated from the naturally variable axillary bud in the *L. indica* ‘Fenjing’ wild-type (WT) variety via asexual cutting propagation [28]. Proteomics studies have shown that the protein abundance of photosynthesis, phenylpropanoid biosynthesis, and phenylalanine metabolism pathways in the *gl1* mutant are significantly different compared to WT [29]. In the *gl1* mutant, photooxidation is a major physiological process, with biological processes adapted for acclimation rather than growth. This results in significant differences in some physiological and biochemical processes, including primary and secondary metabolism [30]. The change in secondary metabolism results in a significantly higher level of phenolic compounds in the *gl1* mutant [30], which is usually a common adaptation of plants in response to stress environments [23,31]. This change provides a stable internal environment for the growth and fitness of the yellow-leaf mutant *gl1*. It remains unclear whether changes in phenylpropanoid metabolism lead to specific bioactive metabolites linked to leaf color variation, as colored grains and fruits rich in phenolic compounds offer increased potential as functional foods [32,33]. Untargeted metabolic profiling involves the comprehensive screening of all metabolites in an assayed sample and allows further exploration of metabolites of interest. Therefore, this study aimed to investigate the phytochemical characteristics of metabolites in the yellow-leaf mutant *gl1* leaves based on chemical and metabolomics approaches. A comparative analysis of the metabolite composition and antioxidant capacity was performed to determine the potential value of the *gl1* mutant as a functional food.

## 2. Materials and Methods

### 2.1. Plant Materials and Chemicals

Six-year-old *gl1* mutant and WT plants, derived from asexual cuttings, were cultivated in the same field at the Institute of Botany, Jiangsu Province, and the Chinese Academy of Sciences (Nanjing, China). The leaves in the same position from at least three different *gl1* mutant or WT plants were collected as individual samples in May 2021 and dried at 70 °C until a constant weight was achieved. Each group of samples was repeated four times as an independent biological replicate. The dry samples were then ground and sifted through a 60-mesh sieve to a fine powder for subsequent use.

Mass spectrometric grade methanol (Merck Co., Darmstadt, Germany) was used for extraction and elution. All other chemicals including Folin-Ciocalteu reagent, ethanol, sodium phosphate bibasic, sodium carbonate, aluminum nitrate, sodium hydroxide, sodium nitrite, potassium persulfate, sodium acetate, ferric chloride (FeCl_3_), ferrous sulfate (FeSO_4_), hydrogen peroxide (H_2_O_2_), hydrochloric acid (HCl), sulfuric acid, ammonium persulfate, salicylic acid, hydroxylamine hydrochloride, sulfanilamide, α-naphthylamine, gallic acid, schizandrol A, rutin, 6-hydroxy-2,5,7,8-tetramethylchroman-2-carboxylic acid (Trolox), 2,2-diphenyl-1-picrylhydrazyl (DPPH), 2,2′-azinobis (3-ethyl-benzothiazoline-6-sulphonic acid ammonium salt) (ABTS), 2,4,6-tripyridyl-s-triazine (TPTZ), and *N*,*N*,*N*′,*N*′-tetramethylethylenediamine (TEMED) were analytical grade (Aladdin Biochemical Technology Co., Ltd., Shanghai, China). Solid-phase extraction (SPE) was performed using Extract Clean^™^ C_18_ ProElut (American Chromatography Supplies, Vineland, NJ, USA).

### 2.2. Determination of Lignin Content

Lignin content was measured using the modified Klason method [34]. Briefly, a total of 0.1 g of ground samples was extracted first with 40 mL of water, 40 mL of 80% ethanol, and then 40 mL of acetone. The samples were then freeze-dried for two days. The dry samples were treated with 3 mL of 72% sulfuric acid for 16 h, stirring every 10 min at room temperature. The samples were then diluted to 4% sulfuric acid with 51 mL of nanopure water. The solutions were transferred to flasks and autoclaved at 121 °C for 60 min. The lignin was collected in preweighted filtering crucibles and washed with 200 mL of nanopure water to remove residual acid before drying at 70 °C to a constant weight.

### 2.3. Histological Analysis

The petiole base, after cutting, was immediately placed in 50 mL of FAA solution (70% ethanol, formaldehyde, and glacial acetic acid in 90:5:5 *v*/*v/v*) for 48 h. The samples were then dehydrated in an ethanol series and transversely sectioned (4 μm) on a Leica RM2016 microtome (Leica, Saarbrücken, Germany). The sections were stained with the Weisner reagent (1% phloroglucinol and 6 M HCl) and immediately observed under a Nikon Eclipse E100 microscope with images captured using a Nikon DS-U3 digital camera (Nikon, Tokyo, Japan). Other sections were stained with 0.1% toluidine blue, and microphotographs were taken with the microscope and digital camera above.

### 2.4. Extraction of Total Soluble Phenolic Compounds

Four individual sample replicates of the *gl1* mutant and WT (0.5 g each) were extracted in 15 mL of 80% methanol (*v*/*v*) with a KQ-250DS ultrasonic apparatus (Kunshan ultrasonic instrument Co., Ltd., Shanghai, China) for 30 min at room temperature. The crude extracts were centrifuged at 10,000× *g* for 10 min at 4 °C. For the quantification of total soluble phenolic compounds, flavonoids, and antioxidant capacity, the extracts were then stored at −20 °C for subsequent analysis. For the qualitative and quantitative analysis of the extracted phenolic compounds, equal amounts of extracts were purified and enriched using SPE C_18_ ProElut according to the manufacturer’s instructions and eluted with 0.5 mL of methanol. The eluents were then collected in an amber vial for future use.

### 2.5. Determination of Total Soluble Phenolic Compounds and Flavonoid Content

Total soluble phenolic content was determined using the Folin-Ciocalteu reagent according to the previously described method [35] with some modifications. Briefly, 150 µL of appropriately diluted extracts or standard solutions were added to 150 µL of Folin-Ciocalteu reagent. The mixture was vortexed and equilibrated for 5 min at room temperature. Then, 200 µL of 1 M sodium carbonate was thoroughly mixed and equilibrated for 10 min at room temperature, with 1 mL of H_2_O added. After a 1-h incubation at room temperature in the dark, the changes in the absorbance of the samples were recorded at 725 nm using a spectrophotometer. The concentrations were calculated according to the calibration curve built using standard solutions of gallic acid. The content was expressed as milligrams of gallic acid equivalents (GAE) in the dry weight of the sample (mg GAE g^−1^ DW).

The flavonoid content was determined using the previously described colorimetric method [36]. Briefly, 5 mL of the extract was mixed with 0.3 mL of 5% sodium nitrite and equilibrated for 5 min at room temperature. Then, 0.3 mL of 10% aluminum nitrate was added and incubated for another 5 min. A total of 4 mL of 1 M sodium hydroxide was added, and the solution was finally filled up to 10 mL with 70% aqueous ethanol. The absorbance of the sample was measured using a spectrophotometer at 510 nm. The content was calculated from a calibration curve using standard rutin solutions and expressed as rutin equivalents (RE) in the dry weight of the sample (mg RE g^−1^ DW).

### 2.6. Antioxidant Activity Assay

The DPPH radical scavenging activity was measured by assessing the ability of antioxidant substances in the extract to capture single electrons from DPPH, reflected as a change in absorbance at 515 nm. These procedures were performed according to a previously described method [37] with minor modifications in the dilution of extracts. Briefly, 2 mL of extraction solution was mixed with 3.5 mL of 0.1 mM DPPH and incubated for 30 min in the dark at 25 °C to determine the absorbance. A standard curve was established using a plot of Trolox concentration against DPPH radical scavenging activity. The DPPH radical scavenging activity values were expressed as micromoles of Trolox equivalent (TE) in the dry weight of the sample (mmol TE 100 g^−1^ DW).

ABTS radical scavenging activity was performed based on the reduction of ABTS^+^ radicals to ABTS by the antioxidant substances in the extract, reflected in the absorbance change of the samples at 734 nm. The preparation and usage of ABTS^+^ radicals working solution were carried out using a previous method [32]. Briefly, the ABTS^+^ stock solution was prepared by mixing 5 mL of 7.4 mM ABTS with 5 mL of 2.6 mM potassium persulfate, followed by overnight incubation at room temperature in the dark. The ABTS^+^ working solution was then diluted with methanol to reach an absorbance of 0.7 ± 0.02 at 734 nm before use. Then, 0.1 mL of extraction solution was mixed with 1.4 mL of ABTS^+^ working solution and incubated for 1 h at room temperature in the dark. The absorbance was read at 734 nm. A series of Trolox standard solutions in 80% methanol was prepared and assayed under the same conditions to produce the standard curve. The ABTS radical scavenging activity values were expressed as mmol TE 100 g^−1^ DW.

The ferric-reducing antioxidant power (FRAP) assay was carried out using the working solution prepared immediately prior to use by mixing 300 mM acetate buffer (pH 3.6), 10 mM TPTZ in 40 mM HCl, and 20 mM FeCl_3_ at a ratio of 10:1:1 (*v*/*v*/*v*) as previously described [32]. FRAP was determined by measuring the change in absorbance at 600 nm. A series of Trolox concentrations were assayed under the same conditions to produce the standard curve. The FRAP of the sample was expressed as mmol TE 100 g^−1^ DW.

Hydroxyl radical scavenging activity was assessed by generating hydroxyl radicals via a Fenton reaction involving equal volumes of 6 mM FeSO_4_ and 24 mM H_2_O_2_. The methodology was carried out according to a previous report [32]. Ultrapure water was used instead of H_2_O_2_ as a control, and the absorbance at 510 nm was A_0_. The absorbance of different diluted extracts at 510 nm was successively recorded as A_1_. The scavenging activity was calculated in percent value as follows: scavenging activity (%) = (A0 − A_1_)/A_0_ × 100.

The superoxide radical (O_2_^•−^) scavenging activity was measured by introducing the superoxide anion to the extracts. The rest of the superoxide anion in the reaction system was determined using the Li et al. [30] method. Briefly, 50 µL of TEMED and 200 µL of 10% ammonium persulfate were thoroughly mixed with different diluted extracts and incubated at room temperature for 1 min. Then, 250 µL of 10 mM hydroxylamine hydrochloride was added to the mixture, vortexed, and incubated at 37 °C for 30 min. Finally, 250 μL of 17 mM sulfanilamide and 250 μL of 7 mM α-naphthylamine were added to the incubation mixture for 20 min to end the reaction. The absorbance of the solution at 530 nm was recorded as A_1_. The absorbance of ultrapure water, serving as a control under the same conditions, was recorded as A_0_. The eliminating rate of superoxide anion was calculated in percent value according to the following formula: scavenging activity (%) = (A_0_ − A_1_)/A_0_ × 100.

### 2.7. Untargeted Metabolomics Analysis

Metabolites were identified using a high-resolution hybrid quadrupole-time-of-flight mass spectrometer (QTOF-MS) with a dual electrospray ionization source coupled to an ultra-high-pressure liquid chromatography system (UHPLC-ESI-QTOF-MS). The 1260 UHPLC chromatograph was equipped with a binary pump and a thermostated column coupled to a G6530A mass spectrometer detector with a MassHunter B0.05.0 workstation (Agilent Technologies, Santa Clara, CA, USA). The mass spectrometer was operated in a negative mode to acquire accurate masses in the 100–1700 *m*/*z* range, and the substance was monitored with dual ESI at 365 and 254 nm. Chromatographic separation was achieved in the reverse phase mode on an Agilent InfinityLab Poroshell 120 SB-AQ column (3.0 mm × 100 mm, i.d. 2.7 μm) under a water-acetonitrile gradient elution. The mobile phases labeled A (0.1% formic acid) and B (acetonitrile) were used with a column temperature maintained at 30 °C. The gradient elution was 90% A and 10% B (5 min), 82% A and 18% B (10 min), 75% A and 25% B (30 min), 10% A and 90% B (35 min), and 100% B (40 min) with a flow rate of 0.4 mL min^−1^. The injection volume was 5 μL, and source conditions for the untargeted negative scan were as follows: nitrogen was used as the drying gas (10 L min^−1^ at 350 °C), the nebulizer pressure was 50 psig, the fragmentor voltage was 130 V, the capillary voltage was 3.5 kV, and the MS/MS collision energy was 25 V.

The raw data were processed using the Profinder 10.0 software (Agilent Technologies, Santa Clara, CA, USA), employing the ‘find-by-formula’ algorithm based on accurate monoisotopic mass (accuracy threshold: 5 ppm) and isotopic pattern (ratio and spacing). The compounds were annotated using the Metlin database Metabolite Link (metlin.scripps.edu). The extracted ion current for the compounds with a score above 70% and a mass accuracy below 5 ppm was used, and the corresponding peak area was exported. Post-acquisition data pre-processing (mass and retention time alignment and compound filtering) was conducted using Mass Profiler Professional 15.1 (Agilent Technologies, Santa Clara, CA, USA). Only those compounds identified within 75% of replications in at least one genotype were used as minimum identification criteria. Finally, this processed dataset was used for further statistical analysis.

### 2.8. Data Analysis

A principal component analysis (PCA) was performed on the unsupervised multivariate metabolomic data to describe the relationships among the variables and the groupings among samples [38]. Compound abundance was normalized at baseline against the median value of all samples. The discriminant metabolites were then subjected to fold change (FC) cut-off analysis, which was Log2 transformed in the Mass Profiler Professional 15.1.

Data are presented as the mean ± standard deviation (SD) from three to four independent biological replicates, statistically analyzed using one-way ANOVA in IBM SPSS 20.0 (IBM, Armonk, NY, USA).

## 3. Results and Discussion

### 3.1. Quantitative Analysis of Phenolic Compounds

Obvious leaf color variation between WT and the *gl1* mutant is due to significant differences in protein expression and physiological processes [29,30]. In the *gl1* mutant, phenylpropanoid biosynthesis and phenylalanine metabolism are key biological processes that significantly differ from WT and are underexplored beyond photosynthesis [29]. Significantly down-regulated proteins in the *gl1* mutant, including Dirigent-like, peroxidase (POD), 4-coumarate: CoA ligase (4CL), cinnamyl alcohol dehydrogenase (CAD), caffeic acid O-methyltransferase (COMT), caffeoyl-CoA O-methyltransferase (CCoAOMT), and shikimate O-hydroxycinnamoyl transferase (HCT) (Appendix A), are directly involved in the pathway from phenylalanine to the monolignols in lignin biosynthesis [39]. Lignin is an integral cell wall constituent of all vascular plants and is a cross-linked phenolic polymer (C6-C3 units) derived from typical polymerization of the hydroxycinnamyl alcohol, *p*-coumaryl, coniferyl, and sinapyl alcohol [39]. Lignin content was therefore determined and a significant decrease in DW was observed in the *gl1* mutant (181.63 ± 14.26 mg g^−1^) compared to WT (223.75 ± 3.28 mg g^−1^) (Figure 1A). Furthermore, phloroglucinol stain, which reacts with conifer aldehyde groups in lignin, demonstrated a significantly lower number of stained xylem cells in the vascular bundles at the base of the mutant petiole (Figure 2A,B). Lignin distribution around the cell wall in the mutant was also irregular and disorderly compared to the WT (Figure 2C,D). Toluidine blue staining revealed similar morphological changes (Figure 2E,F). The decreased lignin content and irregular xylem structure in the *gl1* mutant suggest a lower phenolic polymer lignin level than in WT.

Unlike the phenolic polymer lignin content trend, the soluble phenolic compounds content of methanol extract was significantly higher in the *gl1* mutant (83.33 ± 5.44 mg GAE g^−1^ DW) compared to WT (62.05 ± 1.25 mg GAE g^−1^ DW) (Figure 1B). These results are consistent with the fresh weight results in a previous study [30]. Soluble phenolic compounds, comprising a heterogeneous group of molecules, are typically classified into flavonoids and non-flavonoids, including phenolic acids and lignans [40]. Furthermore, flavonoid content was significantly lower in the mutant leaves (24.87 ± 4.19 mg RE g^−1^ DW) compared to WT (35.17 ± 4.45 mg RE g^−1^ DW) (Figure 1C). These results suggest that the increased soluble phenolic compounds were not flavonoids and the decreased metabolic flux from the lignin did not shift to flavonoid biosynthesis. The influence of yellow-leaf variation on the metabolites of the *gl1* mutant may be linked to non-flavonoid phenolic compounds. The accumulating soluble phenolic compounds in the *gl1* mutant may be pathway intermediates and derivatives, a response to changed system-wide metabolome observed in plants with disturbed lignin biosynthesis [41].

Soluble phenolic compound content at different developmental leaf stages was assayed using the same method to further confirm the differences in soluble phenolic compounds between the *gl1* mutant and WT. The soluble phenolic compound content of immature and mature leaves was 67.98 ± 5.21 and 91.37 ± 1.21 mg GAE g^−1^ DW in the *gl1* mutant, respectively. This was significantly higher than at the same leaf developmental stage in WT (52.39 ± 8.46 and 65.82 ± 6.11 mg GAE g^−1^ DW, respectively) (Appendix A). Furthermore, the soluble phenolic compounds in the mature leaves were significantly higher than in the immature leaves in both mutant and WT plants, consistent with the accumulation of increased metabolites in the mature leaves. The content of phenolic compounds in this study was comparable with previously reported data in *L. speciosa* leaves using the same extraction method, which was 71.06 ± 2.01 mg GAE g^−1^ DW [42] and higher than in the hot water extract of 35.36 ± 2.17 mg GAE g^−1^ DW [43]. The phenolic compound content in *L. speciosa* exceeded that in *Morus alba* (7.11 ± 0.21 mg GAE g^−1^ DW) and *Thunbergia laurifolia* (4.24 ± 0.15 mg GAE g^−1^ DW) [43]. Compared to 26 natural spices and herbs with previously reported antioxidants [44], the phenolic compound content in this study was higher than 18 of them. Therefore, *L. indica* is a plant with high content of soluble phenolic compounds and can be used as a healthy plant product resource, particularly the yellow-leaf *gl1* mutant.

### 3.2. Metabolite Profiling

Systematic metabolomic profiling of the *gl1* mutant and WT leaves was carried out using an untargeted metabolomics approach to investigate the metabolite changes associated with yellow leaves in the *gl1* mutant. Metabolic compounds were eluted via chromatography (Appendix A) and identified by analyzing fragmentation patterns from mass spectra in negative ionization mode. A total of 815 major and minor compounds were identified by matching the mass-to-charge ratios and retention times of peaks in the mass spectrum. Each metabolite was presented by a different response factor at the electrospray ion source exhibiting a specific peak area. The relative quantification of each compound was evaluated by the normalized FC of the peak area present in the mutant and WT. In the mutant-to-WT comparison, 193 differential metabolites, including 84 up-regulated and 109 down-regulated (*p* < 0.05 and FC > 2 or FC < −2), were identified via multivariate statistical analyses, which were classified into 10 known classes (Appendix A). Principal components 1 and 2 accounted for 91.90% and 2.04% of the total variance, respectively, illustrating distinct segregation patterns between the *gl1* mutant and WT (Figure 3A). Among them, 102 differential metabolites exhibited a greater absolute normalization FC value than 4, that is |log_2_ FC| > 2, accounting for 52.85% of differential metabolites, revealing that the metabolite content of the *gl1* mutant was significantly different from those of WT (Figure 3B). Following the merging of the data from the four replicates of each group and considering the RSD of replicates, 56 differential metabolites were finally screened, including 21 phenolic compounds, 15 terpenes, 10 alkaloids (Table 1), and others. Furthermore, the most differential metabolites with |log_2_ FC| > 15 were largely distributed in the phenolic compounds from the point of the comparison of FC in the peak area (bold type lines in Table 1). Coumarinic acid-*β*-d-glucoside, apigenin 7-*O*-glucoside, and 1-*O*,6-*O*-digalloyl-β-d-glucose were detected in the samples of both groups, while myricomplanoside and rhein-8-*O*-*β*-d-(6′-oxalyl)-glucopyranoside were only detected in the WT group, and balanophonin-4-*O*-d-glu and volkensiflavone were detected in higher occurrence frequency in the *gl1* mutant group (Table 1).The difference in metabolite species and content between *gl1* and WT suggested that the leaf color variation of the *gl1* mutant was correlated with the accumulation or production of some specific phenolic compounds.

In the *gl1* mutant, compared to WT, three lignans—balanophonin-4-*O*-d-glu, angeloylgomisin R, and Gomisin B—were identified as differentially up-regulated metabolites. Balanophonin-4-*O*-d-glu specifically demonstrated the highest |log_2_ FC| value among the metabolites (Table 1). Balanophonin is a dihydrobenzo[*b*]furan skeleton neolignan [45] found in different tissues of plants, including the stems [46,47], leaves [48], Hazelnut shells [49], seeds ([50], fruits [51], and callus [52]. The differences in substances downstream of phenylpropane metabolism may be related to the gene or protein expression at key transition points in the pathway. The differentially down-regulated proteins (Dirigent-like, POD, 4CL, CAD, COMT, CCoAOMT, and HCT) in the *gl1* mutant prevented the upstream precursors from forming lignin, resulting in the accumulation of low molecular weight phenolic compounds such as coniferyl and sinapyl alcohols. These alcohols likely form lignans through oxidative coupling, a process wherein lignin (a polymer) and lignan/neolignan (a dimer) are produced by the coupling of C6-C3 phenoxy radicals [53]. Furthermore, the biochemical pathway of lignans is closely related to, although a distinct form of, lignin biosynthesis [53]. Therefore, the phenylpropane metabolism of the *gl1* mutant was hypothesized to be regulated with the decreased metabolic flux from the polymer of phenolic compounds to lignin shifted to the bimolecular oxidative coupling of phenolic compounds to lignans. The accumulation of balanophonin-4-*O*-D-glu in *gl1* confirmed that the lignin flux reductions were associated with the increase in various classes of 4-O- phenylpropanoids [41]. Since the occurrence frequency in WT was only one in four replicates, this suggests that balanophonin-4-*O*-D-glu may be a phenolic compound specific to the *gl1* mutant. The specific metabolites might be the products of the mutant to maintain a dynamic balance for plant growth and fitness by reallocating the metabolic flow between different branches of the phenylpropanoid metabolic pathway and between different sub-branches of the same branch.

### 3.3. Antioxidant Capacity

Antioxidant capacity reflects the ability to scavenge free radicals in food and other biological systems [54]. Phenolic compounds are generally regarded as one of the most desired antioxidants since they are highly active in reducing agents, hydrogen donors, and singlet oxygen quenchers [55]. Consequently, various methods were adopted to assess the antioxidant properties from different perspectives. In the present study, the FRAP and the scavenging activity of DPPH, ABTS, hydroxyl radicals, and superoxide radicals were used to assess the antioxidant capacity of *gl1* and WT leaves. The results demonstrated that the *gl1* mutant exhibited significantly higher FRAP, DPPH, and ABTS values than WT (Figure 4A,B). Contrary to previous reports where FRAP values were higher than DPPH and ABTS [56,57], in *L. indica*, FRAP values were consistently lower than those for ABTS and DPPH. These results suggest that the bioactive compounds in *L. indica* leaves remove free radicals more efficiently compared to reducing ferric ions since FRAP, as measuring reducing power cannot detect compounds that act via radical quenching [58]. The scavenging efficiencies of the extracts differed for hydroxyl and superoxide radicals. The mutant exhibited a significantly higher scavenging efficiency for hydroxyl radicals compared to WT, while no significant differences were observed for superoxide radicals at the same diluted concentration (Figure 4C,D). In a previous report, *gl1* demonstrated an extremely low level of H_2_O_2_ but a notably higher O_2_^•−^ production rate than WT [30].

To better evaluate the antioxidant properties of the *gl1* mutant, statistical analyses were conducted to assess correlations between antioxidant capacity, soluble phenolic compounds, H_2_O_2_, and O_2_^•−^. FRAP, DPPH, ABTS, and hydroxyl radical scavenging activity were positively linked with the soluble phenolic compounds (0.878 to 0.931; *p* < 0.05) (Table 2). Furthermore, a strong negative correlation was identified between FRAP, DPPH, ABTS, hydroxyl radical scavenging activity, and H_2_O_2_ content (−0.931 to −0.996; *p* < 0.01). Interestingly, no correlation was detected between phenolic compounds and superoxide radical scavenging activity (Table 2). While a significant positive correlation was detected between FRAP, DPPH, ABTS, hydroxyl radical scavenging activity, and O_2_^•−^ content (0.829 to 0.971; *p* < 0.05). In plants, phenolic compounds can donate electrons to guaiacol-POD to detoxify H_2_O_2_ produced under stress conditions [31,59]. A significant increase in guaiacol-POD activity in the *gl1* mutant can effectively promote the detoxification of H_2_O_2_ by phenolic compounds, resulting in a substantial decrease in the H_2_O_2_ content in *gl1* leaves [30]. This suggests that phenolic compounds may contribute to the *L. indica*’s antioxidant activity. Taken together, the metabolites of the *gl1* mutant exhibited high antioxidant capacity, selectively scavenging hydroxyl radicals rather than superoxide radicals.

## 4. Conclusions

The *gl1* mutant exhibited an irregular xylem structure, significantly lower lignin content, and the accumulation of phenylpropanoid pathway intermediates and derivatives. Untargeted metabolomics analysis revealed that soluble phenolic compounds were the most significantly differing metabolites between the *gl1* mutant and WT. Lignans, identified as the most differentially up-regulated metabolites, were associated with altered phenylpropanoid metabolism in the *gl1* mutant. The mutant exhibited increased scavenging activity in FRAP, ABTS, DPPH, and hydroxyl radicals compared to WT. Correlation analysis showed that antioxidant capacity was positively correlated with soluble phenolic compounds and negatively correlated with H_2_O_2_ content. This study is the first to specifically address the metabolites associated with leaf color variation in the yellow-leaf *gl1* mutant *L. indica,* highlighting characteristic metabolites with the changes in phenylpropanoid metabolism, which lay a foundation for its putative applications in the food industry.

## Figures and Tables

**Figure 1 plants-13-00315-f001:**
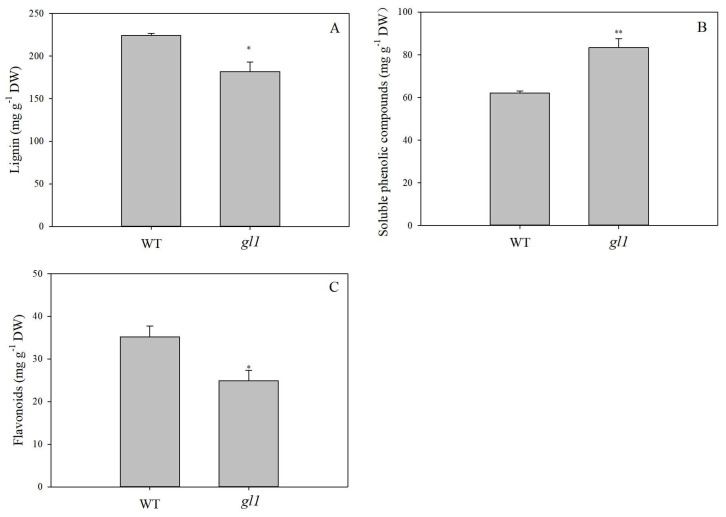
Lignin (**A**), soluble phenolic compounds (**B**), and flavonoid content (**C**) in *gl1* mutant and WT leaves. The vertical bars are presented as mean ± SD, *n* = 3. The asterisks * and ** indicate significant differences at *p* < 0.05 and *p* < 0.01 levels, respectively, between WT and *gl1* mutant.

**Figure 2 plants-13-00315-f002:**
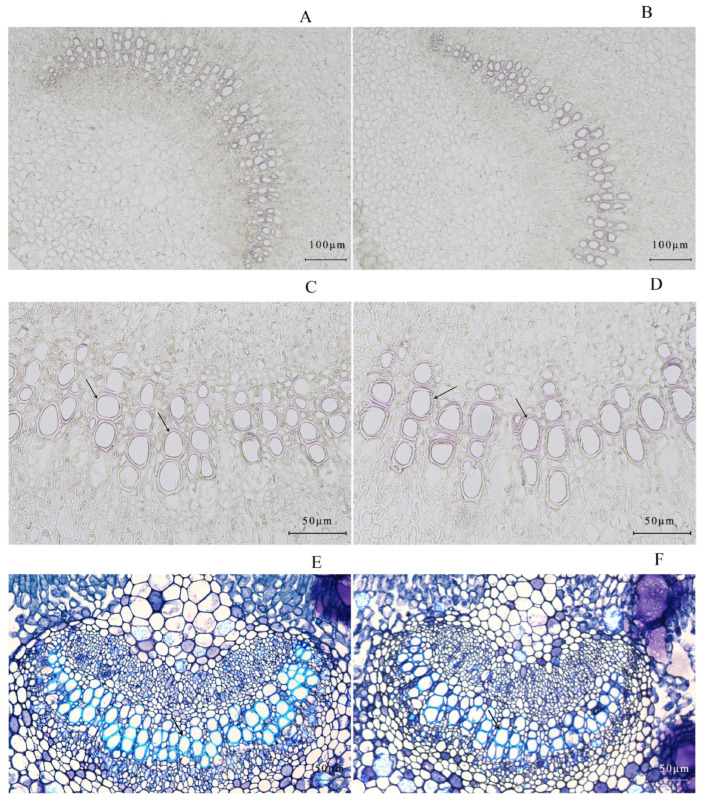
Transverse sections of vascular bundles at the base of the leaf petiole in WT (**A**,**C**,**E**) and *gl1* mutant (**B**,**D**,**F**). Microphotographs of phloroglucinol-HCl staining under 10× objective (**A**,**B**) and 20× objective (**C**,**D**); Microphotographs of toluidine blue staining under 10× objective (**E**,**F**).

**Figure 3 plants-13-00315-f003:**
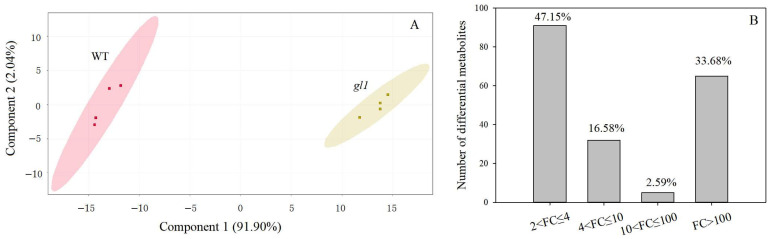
Metabolome analyses of *gl1* mutant and WT leaves. (**A**): PCA plot showing a significant separation of samples from the mutant and WT; (**B**): Number of differential metabolites based on normalization FC value, and the percentage above the vertical bars is its proportion of the total.

**Figure 4 plants-13-00315-f004:**
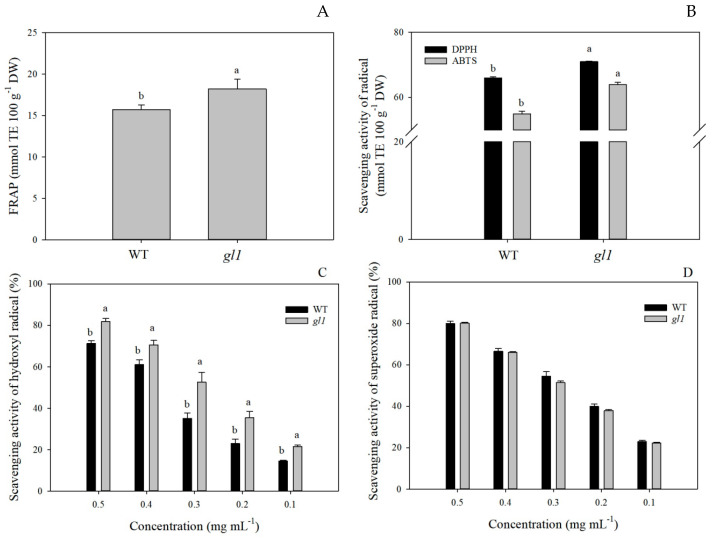
The antioxidant capacity of *gl1* and WT at different concentrations. (**A**) Ferric-reducing antioxidant power (FRAP) assay; (**B**) DPPH and ABTS radical scavenging activity; (**C**) hydroxyl radical scavenging activity; (**D**) superoxide radical scavenging activity; The vertical bars are presented as mean ± SD, n = 4. Different letters represent significant differences at *p* < 0.05 between the *gl1* mutant and WT.

**Table 1 plants-13-00315-t001:** The differential metabolites identified between yellow-leaf *gl1* mutant and WT via the normalized fold change (FC) of peak area and the relative standard deviation (RSD) of the four replicates.

Class/Subclass	Compound	Formula	RT (min)	OF in *gl1*	OF in WT	RSD of *gl1*	RSD of WT	Log_2_ FC	Accumulation
phenolic compounds									
lignans/neolignans									
	**Balanophonin-4-*O*-d-glu**	**C_26_H_30_O_11_**	**1.10**	**4**	**1**	**2.78**	**2.79**	**18.99**	**up**
	Angeloylgomisin r	C_27_H_30_O_8_	39.41	4	4	1.15	3.73	13.96	up
	Gomisin B	C_27_H_30_O_9_	37.06	4	4	5.10	3.90	13.54	up
phenolic acids									
	**1-*O*,6-*O*-Digalloyl-*β*-d-glucose**	**C_20_H_20_O_14_**	**21.21**	**4**	**2**	**1.37**	**3.01**	**17.33**	**up**
	(1*R*,2*S*,4*S*,5*R*,7*R*,9*S*,10*R*)-1-Benzoyloxy-2,15-diacetoxy-4-hydroxy-9-cinnamoyloxy-dihydroagarofuran	C_35_H_40_O_10_	14.17	4	4	5.02	3.64	14.75	up
	Mesuanic acid	C_35_H_48_O_6_	36.55	4	2	1.49	3.86	13.49	up
	Albaspidin ab	C_23_H_28_O_8_	39.94	4	4	3.39	3.72	14.26	up
	**Coumarinic acid-*β*-d-glucoside**	**C_15_H_18_O_8_**	**6.70**	**4**	**4**	**1.40**	**1.75**	**−15.39**	**down**
	Acrovestone	C_32_H_42_O_8_	39.47	4	4	1.54	4.05	−14.32	down
	2′-Hydroxybiphenyl-2-sulfinate	C_12_H_10_O_3_S	1.50	0	4	1.67	3.85	−13.14	down
flavonoids									
	**Volkensiflavone**	**C_30_H_20_O_10_**	**32.81**	**4**	**1**	**5.20**	**3.25**	**16.61**	**up**
	Dihydroisomorellin	C_33_H_38_O_7_	14.67	4	4	4.97	3.89	13.65	up
	Anticancer flavonoid pmv70p691-114	C_19_H_18_O_7_	35.37	4	4	8.83	3.76	14.72	up
	Hispaglabridin A	C_25_H_30_O_4_	38.59	4	4	2.39	3.64	14.38	up
	**Myricomplanoside**	**C_22_H_22_O_13_**	**13.60**	**0**	**4**	**1.37**	**5.48**	**−16.16**	**down**
	**Apigenin 7-*O*-glucoside**	**C_21_H_20_O_10_**	**15.98**	**4**	**4**	**1.46**	**5.42**	**−15.21**	**down**
	Spinoside A	C_39_H_56_O_12_	37.14	3	4	1.49	3.74	−14.61	down
	Kuwanone G	C_40_H_36_O_11_	35.47	4	4	1.64	9.75	−13.60	down
	Gerronemin F	C_28_H_38_O_4_	35.89	4	4	1.57	9.24	−14.16	down
anthraquinone									
	**Rhein-8-*O*-*β*-d-(6′-oxalyl)-glucopyranoside**	**C_23_H_18_O_14_**	**1.15**	**0**	**4**	**1.30**	**1.39**	**−16.57**	**down**
	Morellin	C_33_H_36_O_7_	41.44	3	4	1.59	6.08	−13.83	down
terpenoid									
	Gibberellin A37	C_20_H_28_O_6_	39.95	4	4	4.05	3.77	13.99	up
	**Yunnanxol**	**C_40_H_46_O_14_**	**39.51**	**4**	**4**	**2.86**	**3.33**	**15.88**	**up**
	Taxuspine Q	C_33_H_46_O_13_	39.51	4	4	2.04	3.50	14.95	up
	Isonuezhenide	C_31_H_42_O_17_	39.51	4	4	0.84	3.51	14.81	up
	Austroside a	C_19_H_30_O_9_	38.23	4	3	2.30	3.58	14.70	up
	Cathidin D	C_32_H_37_NO_11_	37.22	4	0	2.08	3.61	14.52	up
	Alatenoside (C-7a-OH epimer)	C_34_H_50_O_21_	39.69	4	4	4.89	3.69	14.44	up
	Dehydroiridodialo-D-gentiobioside	C_22_H_34_O_12_	36.86	4	4	1.67	7.56	−13.45	down
	*trans*-*p*-Hydroxycinnamoylrutaevin	C_35_H_36_O_11_	37.30	4	4	1.66	7.05	−13.27	down
	Acevaltratum	C_24_H_32_O_10_	16.73	4	4	1.65	4.46	−13.18	down
	Paederoside	C_18_H_22_O_10_S	35.06	4	4	1.76	6.66	−12.44	down
	Ailanthone	C_20_H_24_O_7_	14.22	4	4	1.51	5.48	−14.56	down
	10-(*Z*)-*p*-Coumaroyl-6,7-dihydromonotropein	C_25_H_30_O_13_	30.65	4	4	1.52	3.63	−14.32	down
	6-*O*-(4′′-*O*-l-rhamnopy-ranosylvanilloyl)ajugol	C_29_H_40_O_16_	37.30	4	4	1.52	4.21	−14.32	down
	Jatamanvaltrate E	C_28_H_44_O_12_	37.43	4	4	1.55	2.91	−13.96	down
alkaloid									
	Leptopine	C_20_H_18_NO_6_	38.25	4	4	3.12	3.72	14.16	up
	Seneciphylline	C_18_H_23_NO_5_	35.96	4	4	3.89	3.60	14.73	up
	Voacamine	C_43_H_52_N_4_O_5_	37.45	1	4	1.53	4.56	−14.18	down
	Methoxy-5-acetoxy-6-methyl-3-[(*Z*)-10′-pentadecenyl]-1,4-benzoquinone	C_25_H_38_O_5_	38.99	0	4	1.49	5.75	−14.90	down
	Corydamine	C_20_H_18_N_2_O_4_	21.56	4	4	1.54	6.23	−14.48	down
	Aquiledine	C_20_H_20_N_2_O_5_	14.72	4	4	1.62	5.68	−13.63	down
	Gelsamydine	C_29_H_36_N_2_O_6_	39.65	4	4	1.63	4.66	−13.58	down
	Isobetanidin-6-*O*-rhamnosyl sophoroside	C_30_H_36_N_2_O_17_	39.48	4	4	1.64	7.10	−13.54	down
	Protoverine	C_27_H_43_NO_9_	35.77	4	4	1.63	7.20	−13.53	down
	*N*-Methylplatydesmin	C_16_H_20_NO_3_	12.83	4	4	1.77	8.72	−12.91	down

RT: retention time; OF: occurrence frequency; FC: fold change, the normalized peak area ratio of *gl1* mutant/WT groups. The compounds with |log_2_ FC| > 15 are shown in bold type.

**Table 2 plants-13-00315-t002:** Pearson’s correlation of antioxidant capacity and contents of phenolic compounds, H_2_O_2_, and O_2_^•−^ in the mutant and WT.

	Phenolic Compounds	H_2_O_2_	O_2_^•−^
FRAP	0.913 *	−0.931 **	0.829 *
ABTS	0.878 *	−0.961 **	0.971 **
DPPH	0.921 **	−0.975 **	0.937 **
Hydroxyl radical	0.931 **	−0.996 **	0.957 **
Superoxide radical	0.487	−0.569	0.569

* and ** denote significant correlation at *p* < 0.05 and *p* < 0.01, respectively.

## Data Availability

Data is contained within the article.

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
