# Peer review of "Phenolic Compounds and Antioxidant Capacity Comparison of Wild-Type and Yellow-Leaf gl1 Mutant of Lagerstroemia indica"

_plants, 2024, doi:10.3390/plants13020315_

Round 1

Reviewer 1 Report

Comments and Suggestions for Authors

In this manuscript, the authors investigate the phenolic compounds and antioxidant capacity of wild-type and gl1 Lagerstroemia indica, providing valuable insight into the potential applications of this species in the food industry. Clarification of a few points would enhance the manuscript:

Line 252 – please consider replacing “results in significant” with “results from significant,” if this change fits the information given in the sources cited.

Lines 349-350 – do you mean that the leaf color variation “correlated with” the accumulation or production of specific phenolics?

Table 1 – this table would be much easier to read if it were in landscape rather than portrait orientation, as the increased width would prevent words and numbers from being split onto different lines. Please also specify why some compounds and their data are in bold type.

Line 365 – I believe a word other than “shells” would be more appropriate. Also, Esposito et al 2017 does not appear to be in the reference list.

Line 422 – please specify that different letters indicate significant differences between WT and gl1 plants.

Comments on the Quality of English Language

Overall, the English is quite good, though review by a language editor would certainly improve the manuscript.

Author Response

In this manuscript, the authors investigate the phenolic compounds and antioxidant capacity of wild-type and gl1 Lagerstroemia indica, providing valuable insight into the potential applications of this species in the food industry. Clarification of a few points would enhance the manuscript:

  1. Line 252 – please consider replacing "results in significant" with "results from significant," if this change fits the information given in the sources cited.

Thank you for your constructive suggestion. The phenotypic change originates from significant changes in protein expression and physiological processes. We have revised the sentence as per the Reviewer's comments (refer to Lines 251-252 in the revised manuscript).

  1. Lines 349-350 – do you mean that the leaf color variation "correlated with" the accumulation or production of specific phenolics?

Thank you for your feedback. Indeed, the term "correlated with" more accurately conveys the relationship between leaf color variation and the accumulation or production of specific phenolics, as opposed to "resulted in." We have updated the manuscript accordingly to reflect this clarification, in line with the Reviewer's comments. Please refer to Lines 341-344 in the revised manuscript.

  1. Table 1 – this table would be much easier to read if it were in landscape rather than portrait orientation, as the increased width would prevent words and numbers from being split onto different lines. Please also specify why some compounds and their data are in bold type.

Thank you for your valuable input. We agree that presenting Table 1 in a landscape format will enhance readability by preventing the splitting of words and numbers. This adjustment has been made in the revised manuscript. The compounds highlighted in bold type are the differential metabolites with |log2 FC| > 15, indicating their significant difference. As per the Reviewer's comments, we have incorporated these details in both the manuscript text and the table caption for clarity. Please refer to Lines 333-337 and 353 in the revised manuscript for these changes.

  1. Line 365 – I believe a word other than "shells" would be more appropriate. Also, Esposito et al 2017 does not appear to be in the reference list.

Thank you for pointing out the oversight. We have now corrected the terminology to "Hazelnut (Corylus avellana L.) shells," which more accurately describes the material referenced. The missing reference, Esposito et al. 2017, has also been duly added to the reference list. These corrections have been implemented in the manuscript as per the Reviewer's comments. Please refer to Lines 358-360 in the revised manuscript for the updated text and references.

  1. Line 422 – please specify that different letters indicate significant differences between WT and gl1 plants.

Thank you for your suggestion. We have clarified in the text that different letters indicate significant differences between WT and gl1 plants. This correction has been made in line with the Reviewer's comment. Please refer to Lines 421-422 in the revised manuscript.

Reviewer 2 Report

Comments and Suggestions for Authors

Phenolic compounds and antioxidant capacity comparison of wild-type and yellow-leaf gl1 of Lagerstroemia indica

Sumei Li et al

Abstract

Background: The yellow-leaf gl1 mutant of Lagerstroemia indica exhibits an altered phenylpropanoid metabolism pathway compared to wild-type (WT). However, detailed information on the metabolites associated with leaf color variation including the putative production of color-specific metabolites with bioactive constituents remains unknown. Methods: Chemical and metabolomics approaches were used to compare metabolite composition and antioxidant capacity between the gl1 mutant and WT leaves. Results: The mutant exhibited an irregular xylem structure with a significantly lower phenolic polymer lignin content and higher soluble phenolic compounds. Phenolic compounds were the key differential metabolites between gl1 and WT through untargeted metabolomics analysis, especially a significant increase in the type and content of lignans in the mutant. A neoligan derivative balanophonin-4-O-D-glu was demonstrated to be a characteristic metabolite in gl1 mutant. The soluble phenolic compounds of gl1 mutant exhibited higher FRAP, ABTS, DPPH, and hydroxyl radical scavenging activity than in WT. Correlation analysis demonstrated that the antioxidant capacity was positively related to phenolic compounds in L. indica. Conclusions: The metabolites associated with leaf color variation in L. indica yellow-leaf gl1 mutant exhibited high antioxidant capacity, selectively scavenging hydroxyl radicals.

Comment to authors:

In general, this manuscript has been carefully prepared, is logically structured, and can provide noteworthy information for readers. The introduction section is well-developed. However, I believe there is still room for improvement before it can be accepted.  I recommend a minor revision for this manuscript.

Please provide any files of LC-MS detected for Table 1 in the supplementary file. Is it difficult to identify secondary metabolites without HR UPLC-MS?

For table S2, I recommend providing an additional column named "Error Mass" with the calculation of compound errors.

The names of compounds in Table 1 were not prepared carefully following IUPAC. Please check them. Also, the names of compounds in the entire manuscript, such as lines 344-347, also need correction.

The conclusion and discussion are both lacking and of poor quality. Please improve them.

Comments on the Quality of English Language

Moderate editing of English language required

Author Response

In general, this manuscript has been carefully prepared, is logically structured, and can provide noteworthy information for readers. The introduction section is well-developed. However, I believe there is still room for improvement before it can be accepted.  I recommend a minor revision for this manuscript.

  1. Please provide any files of LC-MS detected for Table 1 in the supplementary file. Is it difficult to identify secondary metabolites without HR UPLC-MS?

Thank you for your valuable suggestion. In response, we have included the total ion chromatogram files of eight samples as supplementary material in the revised manuscript. High-resolution untargeted UPLC-MS is indeed instrumental in screening metabolites of interest, facilitating the identification of specific secondary metabolites.

  1. For table S2, I recommend providing an additional column named "Error Mass" with the calculation of compound errors.

Thank you for your suggestion. As per the Reviewer's comments, we have incorporated an additional column titled "Error Mass" in Table S2 of the revised manuscript.

  1. The names of compounds in Table 1 were not prepared carefully following IUPAC. Please check them. Also, the names of compounds in the entire manuscript, such as lines 344-347, also need correction.

Thank you for your insightful comments. We have meticulously reviewed and corrected the names of compounds in Table 1 and throughout the manuscript to align with IUPAC standards. These revisions have been made as per the Reviewer's comments. Please refer to Table 1 and Lines 336-340 in the revised manuscript.

  1. The conclusion and discussion are both lacking and of poor quality. Please improve them.

Thank you for your feedback. Acknowledging the importance of a robust discussion and conclusion, we have thoroughly revised these sections to enhance their quality and depth. These improvements have been incorporated into the revised manuscript. Please refer to the respective sections for the updated content.

Reviewer 3 Report

Comments and Suggestions for Authors

The authors are unable to address my concerns sufficiently to make this manuscript suitable for publication through author revisions.

First, I found significant flaws in the concept and research topic. For example, the research title suggests a focus on metabolite and antioxidant activity, however, there are proteomics-related content in both the introduction and section 3.1. If the proteomics results are important enough to be discussed in the first paragraph, Table S1 should appear in the manuscript instead of the Supplementary File.

The most concerning part of this manuscript is the analytical method validation. In the metabolomics (& proteomics) field, the validation of analytical methods is the most critical part. Even proteomics methods (e.g., Table S1) are missing in 2. materials and methods. The level of annotated chemicals must be present in the supplementary file with proper identifying criteria, including standard chemicals, TIC, mass spectrum, etc. (Sumner, Lloyd W., et al. "Proposed minimum reporting standards for chemical analysis: chemical analysis working group (CAWG) metabolomics standards initiative (MSI)." Metabolomics 3 (2007): 211-221.). In line 323, if 815 are identified compounds, rather than peaks, they may need to be listed in a supplementary file. Once a standard analytical method has been validated, the data analysis can be further achieved and interpreted for drawing results and conclusions.

For statistical analysis, PCA is an unsupervised method and is not able to reveal significant metabolites contributing to differences in cluster patterns between treatments (Line 330-333).

Also, there are some topic keywords/points, such as ‘yellow-leaf gl1 mutant’, ‘mature’, ‘immature’, ‘phenolic compounds’, ‘color’, and ‘phenolic compounds’, and their relations are not sufficient to explain why yellow-leaf gl1 mutant should be studied. It is difficult to understand the highlighted significance like, ‘regardless of the sample drying methodology (Line 305-311)’, ‘good food resource of soluble phenolic compounds……mutant (Line 312-315)’, ‘color variation (even not shown) (Line 453)’, and ‘which laid a foundation for its putative applications in the food industry (Line 454-456)’ after all these experiments.

There are too many vague words, such as ‘same’, ‘differential metabolites’, and ‘previous’. Instead of directly adding references to authors’ previous works such as in bioactive assays and discussion, the appropriate description is necessary to be mentioned in this manuscript (e.g., Line 169-183, 252-256). Also, ‘conventional phenolic compound (Line 343)’ and ‘some new phenolic compounds (Line 346) require clarification. If some are found and reported from this plant material for the first time, further confirmation based on NMR, chemical standard, and/or presenting mass spectra similarity with raw data comparison would be required.

About the figure and table, Fig. 1 missed A, B, and C. Arrows may need to be present in Fig. 2. What does Fig. 3B imply? In general, all titles require improved clarification (sample, treatment, applied statistical method, and comparing traits, etc.).

In addition, there appears to be a need for improvement and clarification to accentuate the findings. During the discussion, certain references appeared inadequate and irrelevant to support the findings. In particular, interpreting data and drawing conclusions must be derived through sequential discussion rather than abrupt transitions (e.g., Lines 271-273, 290-294).

Therefore, the current manuscript format is not acceptable. Authors must thoroughly reconstruct the concept and rationale of this study and resubmit. The logic and correlation between analytical method validation, statistic method, software, interpreting results, selecting proper references, and drawing conclusions, proper discussion derived from the relevant references must be developed with the improvement of scientific writing.

Comments on the Quality of English Language

Additional attention may be required to improve the clarification.

Author Response

The authors are unable to address my concerns sufficiently to make this manuscript suitable for publication through author revisions.

  1. First, I found significant flaws in the concept and research topic. For example, the research title suggests a focus on metabolite and antioxidant activity, however, there are proteomics-related content in both the introduction and section 3.1. If the proteomics results are important enough to be discussed in the first paragraph, Table S1 should appear in the manuscript instead of the Supplementary File.

Thank you for your valuable feedback. We appreciate your insight regarding the incorporation of proteomics content in our manuscript, which primarily focuses on metabolite and antioxidant activity. This research is indeed a continuation of our studies on the yellow-leaf gl1 mutant. The introduction includes proteomics-related findings and other physiological aspects to provide comprehensive background information. The differential expression proteins of the phenylpropanoid metabolite pathway, which are pivotal for understanding changes in secondary metabolites, are detailed in the Supplementary File. This approach aligns the manuscript's focus while offering detailed insights into aspects directly related to lignin biosynthesis.

  1. The most concerning part of this manuscript is the analytical method validation. In the metabolomics (& proteomics) field, the validation of analytical methods is the most critical part. Even proteomics methods (e.g., Table S1) are missing in 2. materials and methods. The level of annotated chemicals must be present in the supplementary file with proper identifying criteria, including standard chemicals, TIC, mass spectrum, etc. (Sumner, Lloyd W., et al. "Proposed minimum reporting standards for chemical analysis: chemical analysis working group (CAWG) metabolomics standards initiative (MSI)." Metabolomics 3 (2007): 211-221.). In line 323, if 815 are identified compounds, rather than peaks, they may need to be listed in a supplementary file. Once a standard analytical method has been validated, the data analysis can be further achieved and interpreted for drawing results and conclusions.

Thank you for your insightful feedback regarding the analytical method validation in our study. In light of your comments, we have made significant enhancements to our manuscript. Firstly, recognizing the importance of proteomics in our research, we have included detailed proteomics results from our prior studies in the Supplementary Materials. This addition provides a comprehensive background and supports the findings presented in our current research. Additionally, we have addressed the critical aspect of method validation by incorporating the Total Ion Chromatograms (TIC) of the eight samples into the Supplementary Files of the revised manuscript. These samples were analyzed using a dual electrospray ionization (ESI) mass spectrometer in negative mode, ensuring precise mass measurements within the 100-1,700 m/z range. This consistent approach across all samples allows for a uniform and comparative analysis.

Furthermore, from the initial identification of 815 compounds, we have focused our analysis on 193 compounds that showed significant differences between the gl1 mutant and the WT. This selection was based on stringent criteria, including p < 0.05 and FC > 2 or FC < -2. These compounds were then subjected to further detailed examination. Your suggestions have been invaluable in reinforcing the scientific rigor of our manuscript. We have updated the relevant sections and the Supplementary Files to reflect these comprehensive changes and improvements.

  1. For statistical analysis, PCA is an unsupervised method and is not able to reveal significant metabolites contributing to differences in cluster patterns between treatments (Line 330-333).

Thank you for your valuable input regarding the use of PCA in our statistical analysis. We agree that the initial description of PCA as a tool for revealing significant metabolite differences between the gl1 mutant and WT was inaccurate. PCA is indeed an unsupervised method that primarily helps in understanding the relationships among variables and the patterns of grouping among samples. In response to your comment, we have revised the relevant section to represent the application and utility of PCA in our analysis correctly. This correction aligns the manuscript more accurately with the standard analytical practices in the field. Please refer to the revised lines (Lines 326-331) in the manuscript for the updated content.

  1. Also, there are some topic keywords/points, such as 'yellow-leaf gl1 mutant', 'mature', 'immature', 'phenolic compounds', 'color', and 'phenolic compounds', and their relations are not sufficient to explain why yellow-leaf gl1 mutant should be studied. It is difficult to understand the highlighted significance like, 'regardless of the sample drying methodology (Line 305-311)', 'good food resource of soluble phenolic compounds……mutant (Line 312-315)', 'color variation (even not shown) (Line 453)', and 'which laid a foundation for its putative applications in the food industry (Line 454-456)' after all these experiments.

Thank you for your constructive feedback. We understand the need to better articulate the significance of studying the yellow-leaf gl1 mutant and its implications. The yellow-leaf gl1 mutant, previously detailed in our earlier reports, has been developed into a colorful ornamental plant variety due to its bright leaf color. However, its practical application has revealed several challenges. The notable leaf color variation between WT and the gl1 mutant is associated with significant differences in internal physiological processes, particularly in phenylpropanoid metabolism. This pathway is crucial due to its wide range of physiological roles in plants and the extensive diversity of its metabolites, exceeding 8,000 in number with varied structural forms (Dong and Lin, 2021). The contrasting directions in the contents of phenolic polymers, such as lignin and soluble phenolic substances in the gl1 mutant, piqued our interest. Our study aims to explore whether the adjustments in phenylpropanoid metabolism lead to the production of specific phenolic compounds that may offer health benefits. This exploration is driven by the hypothesis that the gl1 mutant may synthesize unique phenolic compounds due to its altered metabolic processes. The assessment of soluble phenolic compounds in both mature and immature leaves, along with comparative analyses with other plants, underscores our objective to confirm a higher level of soluble phenolic compounds in the gl1 mutant.

Our research, therefore, not only adds to the existing knowledge of phenylpropanoid metabolism in plant variants but also highlights the potential of the gl1 mutant as a resource for health-promoting compounds.

  1. There are too many vague words, such as 'same', 'differential metabolites', and 'previous'. Instead of directly adding references to authors' previous works such as in bioactive assays and discussion, the appropriate description is necessary to be mentioned in this manuscript (e.g., Line 169-183, 252-256). Also, 'conventional phenolic compound (Line 343)' and 'some new phenolic compounds (Line 346) require clarification. If some are found and reported from this plant material for the first time, further confirmation based on NMR, chemical standard, and/or presenting mass spectra similarity with raw data comparison would be required.

Thank you for your insightful feedback, which has been instrumental in refining our manuscript. Recognizing the need for clarity and precision, we have revised the manuscript, and terms like 'same,' 'differential metabolites,' and 'previous' have been replaced with more specific descriptions, ensuring a clearer and more comprehensive presentation of our findings. Additionally, according to the Reviewer's comments, we have made corrections in describing the bioactive assays and discussing the results as much as possible.

Regarding the clarification of phenolic compounds, we have revised terms such as "conventional phenolic compounds" and "new phenolic compounds" to be more explicit. We now provide clear descriptions of these compounds, avoiding generalizations. Furthermore, while untargeted metabolomics in this study served as an initial screening tool, we plan to undertake more in-depth identification and validation of certain interesting compounds, especially those newly discovered, in future studies using methods like NMR.

These modifications have been made to enhance the manuscript's clarity, accuracy, and overall scientific rigor. We invite you to review the updated sections in the revised manuscript for a detailed understanding of these changes.

Top of Form

  1. About the figure and table, Fig. 1 missed A, B, and C. Arrows may need to be present in Fig. 2. What does Fig. 3B imply? In general, all titles require improved clarification (sample, treatment, applied statistical method, and comparing traits, etc.).

Thank you for pointing out the shortcomings in our figures and tables. We acknowledge the need for more detailed annotations and have addressed these issues in the revised manuscript. Figure 3B is intended to show the proportion of differential metabolites at different thresholds based on their normalized FC values. We have made corrections according to the Reviewer's comments.

  1. In addition, there appears to be a need for improvement and clarification to accentuate the findings. During the discussion, certain references appeared inadequate and irrelevant to support the findings. In particular, interpreting data and drawing conclusions must be derived through sequential discussion rather than abrupt transitions (e.g., Lines 271-273, 290-294).

Thank you for your constructive suggestions regarding the need for improvement and clarification in our manuscript. We have carefully reviewed the discussion section and made significant revisions to better highlight and support our findings. In particular, we have focused on ensuring that the references used are both adequate and relevant, directly supporting the conclusions we draw from our data. See Lines 310-312 and Lines 337-344 in the revised manuscript.

Therefore, the current manuscript format is not acceptable. Authors must thoroughly reconstruct the concept and rationale of this study and resubmit. The logic and correlation between analytical method validation, statistic method, software, interpreting results, selecting proper references, and drawing conclusions, proper discussion derived from the relevant references must be developed with the improvement of scientific writing.

Thank you for your valuable feedback. We have made comprehensive revisions to meet the high standards expected of scientific research publications. Thank you again for your guidance and insights.

Reviewer 4 Report

Comments and Suggestions for Authors

Summary

In this paper, the authors used a chemical and metabolomics approach to compare metabolite composition and antioxidant capacity between the gl1 mutant and WT leaves of Lagerstroemia indica. The results showed that the mutant species exhibited an irregular xylem structure with a significantly lower phenolic polymer lignin content and higher soluble phenolic compounds. Metabolomics analysis showed a significant increase in the type and content of lignans in the mutant. Mutant gl1 was shown to have the characteristic metabolite balanophonin-4-O-D-glu and the soluble phenolic compounds with higher FRAP, ABTS, DPPH, and hydroxyl radical scavenging activity. In conclusion, the authors find that the metabolites associated with leaf color variation in the L. indica yellow-leaf gl1 mutant exhibited high antioxidant capacity, selectively scavenging hydroxyl radicals.

Specific remarks

- The Introduction section is well presented

- The Materials and Methods section is well presented.

- The Results and Discussion section is well presented.

- Conclusions are well presented.

Considering the aforementioned remarks, I recommend this manuscript be published in Plants Journal.

Author Response

Thank you very much!

Round 2

Reviewer 2 Report

Comments and Suggestions for Authors

I am satisfied with this revision. Therefore, I recommend accepting this current form.